# Financial Support for Agriculture, Chemical Fertilizer Use, and Carbon Emissions from Agricultural Production in China

**DOI:** 10.3390/ijerph19127155

**Published:** 2022-06-10

**Authors:** Lili Guo, Sihang Guo, Mengqian Tang, Mengying Su, Houjian Li

**Affiliations:** 1College of Economics, Sichuan Agricultural University, Chengdu 611130, China; 14453@sicau.edu.cn (L.G.); 201907292@stu.sicau.edu.cn (S.G.); tangmengqian@stu.sicau.edu.cn (M.T.); 2College of Economics, Guangxi Minzu University, Nanning 530006, China

**Keywords:** financial support for agriculture, chemical fertilizer use, carbon emissions, agricultural production

## Abstract

In the past 15 years, China has emitted the most carbon dioxide globally. The overuse of chemical fertilizer is an essential reason for agricultural carbon emissions. In recent years, China has paid more and more attention to financial support for agriculture. Therefore, understanding the relationship between chemical fertilizer use, financial support for agriculture, and agricultural carbon emissions will benefit sustainable agricultural production. To achieve the goal of our research, we selected the panel data of 30 provinces (cities) in China from 2000 to 2019 and employed a series of methods in this research. The results demonstrate that: the effect of chemical fertilizer consumption on agricultural carbon emissions is positive. Moreover, financial support for agriculture has a significantly positive impact on reducing carbon emissions from agricultural production. In addition, the results of causality tests testify to one−way causality from financial support for agriculture to carbon emissions from agricultural production, the bidirectional causal relationship between chemical fertilizer use and financial support for agriculture, and two−way causality between chemical fertilizer use and agricultural carbon emissions. Furthermore, the results of variance decomposition analysis represent that financial support for agriculture will significantly affect chemical fertilizer use and carbon emissions in the agricultural sector over the next decade. Finally, we provide several policy suggestions to promote low−carbon agricultural production based on the results of this study. The government should uphold the concept of sustainable agriculture, increase financial support for environmental−friendly agriculture, and encourage the research and use of cleaner agricultural production technologies and chemical fertilizer substitutes.

## 1. Introduction

Global warming, a primary environmental issue, has been widely concerned. Over the past century, the global average temperature has increased by about 1 °C [1]. And excessive greenhouse gas (GHG) emissions are the primary “culprit”. Therefore, laws and regulations, such as the Kyoto Protocol, have been promulgated continuously to meet climate change.

To better understand global environmental issue, many scholars are committed to the research on environmental issue and made many important contributions, such as the environmental Kuznets curve (EKC) hypothesis. The EKC hypothesis shows that when a country’s economic development level is low, environmental pollution is not serious, with the increase of per capita income, environmental pollution tends from low to high, but decreases per capita income as increases further. The economic level of developing countries is low. Moreover, economic growth is preferred than environmental protection in developing countries [2]. Therefore, environmental pollution in developing countries is a major issue in the world. China is the worlds’ largest developing country with the largest population. Actually, in the past 15 years, China has emitted the most carbon dioxide globally [3]. Therefore, China can be the focus of research on environmental pollution in developing countries. At the Leaders’ Climate Summit, China’s government once again pledged that China will strive to achieve the goal of carbon peaking by 2030 and carbon neutrality by 2060. It reflected China’s determination to meet global warming. Agriculture is one of the pillars of China’s national economy. China has to carry out plenty of agricultural activities to feed nearly a quarter of the global population, with less than 10% of global arable land [4]. Unfortunately, agricultural production has consumed a lot of agricultural materials, such as fertilizers, fuels, pesticides, and feeds, which are essential sources of carbon emissions from agricultural production. Moreover, about 20% of China’s total greenhouse gas emissions came from agricultural activities [5]. Therefore, China must reduce carbon emissions, especially in agriculture, as soon as possible.

In recent years, China has further promoted rural revitalization based on building a moderately prosperous society in an all−around way. Agriculture has a more significant effect on reducing poverty than others in less developed areas [6]. Therefore, China’s agriculture must develop rapidly. However, with the rapid development of agriculture, some new situations have occurred. On the one hand, increasing agricultural production activities will influence chemical fertilizer use. With limited resources, lack of cultivated land, and rapid population growth, chemical fertilizer, as an agricultural material that helps to improve yield, has become an important measure to alleviate the pressure on agriculture [7]. Therefore, China has become the largest user of fertilizers, consuming nearly 40% of the world’s fertilizers [8,9]. At the same time, the overuse of chemical fertilizer led to an increase in carbon emissions from agricultural production [10].

On the other hand, the government will provide financial support for agricultural activities. Since 2004, China’s No. 1 central document has been subject to agriculture, rural areas, and farmers. Furthermore, China’s government has paid increasing attention to financial support for agriculture. In such a complicated situation, the impact of agricultural production on the environment has gradually become a crucial issue. Therefore, researching the relationship between chemical fertilizer use, financial support for agriculture, and carbon emissions in the agricultural sector provides a reference and basis for the environmental protection department to formulate policies to promote low−carbon agriculture.

Based on the above, we attempt to use the relevant data in China Rural Statistical Yearbook, China Statistical Yearbook, the China National Bureau of Statistics, and China Population and Employment Statistics Yearbook from 2000 to 2019 to measure annual agricultural carbon emissions of 31 provinces (cities) in China and investigate the relationship between chemical fertilizer use, financial support for agriculture, and carbon emissions from agricultural production. Compared with the existing literature, the three main contributions of this paper are reflected as follows. First, we filled the gap in China’s provincial carbon emissions data from agricultural production by estimating the carbon emissions from agricultural production of 31 provinces (cities) in China. Second, we used some dynamic estimation methods to understand the relationship between our interested variables better. Third, in terms of theoretical significance, this article proved that financial support for agriculture significantly affected carbon emissions from agricultural production and explained its mechanism. Furthermore, in terms of practical significance, this work provided empirical evidence for formulating policies to promote low−carbon agriculture globally, especially in less developed areas.

The rest part of this study is divided into the following four sections. Section 2 is a brief literature review in this field; Section 3 introduces the data source, variables setting, and methodology; Section 4 shows and discusses the empirical results; Section 5 summarizes the previous sections and provides discussion and policy recommendations based on the conclusion and previous research, shortages, and suggestions for future work.

## 2. Literature Review

Nowadays, agricultural production accounts for 10–12% of GHG emissions [11]. Therefore, an increasing number of scholars have concentrated on the significant issue of agricultural carbon emissions. Numerous scholars have estimated carbon emissions from agricultural activities in the past two decades. Tian et al. [12] identified more than 20 kinds of carbon sources and estimated carbon emissions from 1995 to 2010 in China’s agricultural sector. Liu and Gao [13] adopted the MinDW to assess the agricultural carbon emission performance in 11 Changjiang Economic Corridor (CEC) provinces. These researches provided estimates that could quantify carbon emissions from agricultural activities. Moreover, some scholars investigated the factors affecting carbon emissions from agricultural production based on the estimates. From the macroscopic aspect, relative researches are comprehensive, and the influencing factors of carbon emissions from agricultural activities mainly include economic development [14,15], agricultural land use [16,17], technological progress [18,19], agricultural structure adjustment [20], agricultural capital input [10,21] and agricultural production efficiency [22]. From the micro aspect, researchers mainly focused on individual farmers. Jiang et al. [23] pointed out that farmers’ willingness to use biomass waste was helping to reduce carbon emissions in agricultural production. Kipling et al. [24] found that farmers’ poverty in knowledge was one of the main reasons that prevent farmers from taking pro−climate actions. Guan et al. [25] indicated that farmers usually refuse to produce in an environmental−friendly way because of various constraints, such as cost and labor. They proved that the individual behavior of farmers was significant for promoting low−carbon agriculture from the aspects of farmers’ psychology, cognitive ability, and realistic constraints, respectively. Given these influencing factors, numerous scholars actively explore ways to reduce carbon emissions from agriculture. Koondhar et al. [26] focused on Pakistan’s agricultural sector. They found that the farmers in Pakistan need to convert from chemical fertilizer and high−carbon energy consumption to organic fertilizer and clean energy consumption to ensure sustainable cereal food production in Pakistan. Moreover, Liu et al. [27] found that compared with chemical fertilizer, organic fertilizer had better protection for the environment without crop yield loss by experimenting in eastern rural China. They all suggested replacing high−carbon agricultural materials with environmentally−friendly agricultural materials in agricultural production. However, previous studies rarely included financial support for agriculture in discussing agricultural carbon emissions reduction measures.

The previous literature on financial support for agriculture is comprehensive. Fan et al. [28] researched the relationship between the scale and structure of government spending, agricultural productivity, and rural poverty in India. They suggested that the Indian government spend more on rural roads and agricultural research to reduce rural poverty. Tang et al. [29] found that financial support for agriculture has a positive effect on lowering the urban−rural income gap in a significant way. Rada and Valdes [30] found that agricultural infrastructure should receive more financial support. The research about financial support for agriculture mainly focused on its efficiency, scale, structure, and effects on agricultural economic development. Unfortunately, relative fields lack research on the relationship between financial support for agriculture and agriculture carbon emissions [31,32,33,34].

However, many scholars have found that financial support for agriculture significantly affected the influencing factors of agricultural carbon emissions. Firstly, financial support for agriculture influences fertilizer use. Some scholars approved the positive reduction effect of financial support for agriculture on chemical fertilizer use. Fan et al. [35] found that financial support for agriculture would promote the adoption of some new technologies, which would increase fertilizer utilization efficiency [36], and farmers may reduce chemical fertilizer consumption. Wang et al. [37] found that financial support for agriculture encouraged farmers to use organic fertilizer instead of chemical fertilizer. Guo et al. [38] found that financial support for agriculture would reduce chemical fertilizer consumption by adopting technology and scale−up. However, some scholars refuted their opinions. Scholz and Geissler [39] represented that financial support for agriculture would increase farmers’ chemical fertilizer use. Furthermore, Vercammen [40] found that financial support for agriculture is beneficial to farmers to ease financial constraints. And it may cause farmers to buy more chemical fertilizers. Secondly, financial support for agriculture influences agricultural machinery use. Financial support for agriculture would increase the use of machinery [41]. And it would lead to low energy−environment performance [42]. Thirdly, financial support for agriculture changes the scale and planting structure of arable land. Yi et al. [43] found that farmers would like to adjust their arable land’s scale and planting structure to get more financial support related to the scale and planting structure of arable land.

From the existing research, we note that numerous documents have focused on the impacts of finance on the environment and fertilizer use, and fertilizer use on the environment. However, little research discussed the effect of finance on the environment, especially carbon emissions, from the agricultural level. Moreover, most scholars paid attention to the environmental impact of part of the financial support for agriculture (such as agricultural subsidy) on the environment. They seldom discussed the influence of financial support for agriculture on agricultural carbon emissions at an overall level. Compared with the existing research, this study discusses the effect of financial support for agriculture on environment at an overall level, fills the research gap in financial support for agriculture’s environmental effect, and riches the research on carbon emissions, especially in agriculture.

## 3. Materials and Methods

### 3.1. Estimation of Carbon Emissions from Agricultural Production

Agricultural carbon sources mainly involve fossil fuel, fertilizer consumption, deforestation, burning, animal digestion and excretion, and arable land [44]. The agricultural material input in agricultural production is mainly concerned in this study. Therefore, we use chemical fertilizer, pesticides, mulches, diesel, agricultural plowing, agricultural irrigation, agricultural electricity, and livestock as carbon sources.

The carbon emissions measurement formula is as follows:(1)$$C=\sum^{\;}C_{i}=\sum^{\;}T_{i}\cdot \delta_{i}\;$$

Among them, C denotes total carbon emissions from agricultural production; $C_{i}$ represents the carbon emissions from the i−th source. $T_{i}$ illustrates the used amount of the i−th source, and $\delta_{i}$ expresses the carbon emissions coefficient of the i−th source. We summarize each carbon source emission coefficient and their references in Table 1 for readers to read.

### 3.2. Data and Variables

We use annual data of 30 provinces (cities) in China from 2000 to 2019 for empirical analysis. The data are collected from the China Rural Statistical Yearbook (2001–2020), China Statistical Yearbook (2001–2020), the China National Bureau of Statistics, and the China Population and Employment Statistics Yearbook (2001–2020). This study sets the following variables: chemical fertilizer use (perfertilizer), agricultural carbon emissions (percarbon), and financial support for agriculture (agriratio). To mitigate heteroscedasticity, we process the data logarithmically for the analysis. Moreover, we summarize the definition and measurement of each variable in Table 2 for readers to read.

### 3.3. Econometric Model

The primary purpose of our study is to contribute to reducing agricultural carbon emissions. Therefore, we employ the econometric model to explain the relationship between chemical fertilizer use, carbon emissions from agricultural production, and financial support for agriculture. The linear relationship between these variables can be expressed as follows:(2)$$\begin{array}{c} \mathrm{lnpercarbon}=f\left(\mathrm{perfertilizer},\;\mathrm{lnagriratio}\right) \end{array}$$

The model used in this study can be expressed as follows:(3)$$\begin{array}{c} {\mathrm{lnpercarbon}}_{i,t}{=c}_{1}{+\beta }_{1}{\mathrm{lnperfertilizer}}_{i,t}{+\beta }_{2}{\mathrm{lnagriratio}}_{i,t}+\mu \end{array}$$
where $c_{1}$ represents the intercept term; μ denotes the error term; $\beta_{1}$ and $\beta_{2}$ illustrate the coefficients of lnperfertilizer and lnagriratio, respectively.

### 3.4. Cross−Sectional Dependence Test

In the panel data model analysis, the cross−sectional correlation will lead to correlation or cross−sectional heterogeneity between disturbance terms. They may distort estimation [47,48]. To investigate the existence of cross−sectional correlation, we decided to employ the Breusch−Pagan LM test [49], Pesaran Cross−sectional Dependence (CD) test, and Pesaran scaled Lagrange Multiplier (LM) test [50] in the initial stage of this study. Proof of cross−sectional dependence test is shown in Appendix A.

### 3.5. Panel Unit Root Tests

Spurious regression leads to incorrect estimates and will most likely indicate a nonexisting relationships. The panel unit root test is necessary for the panel to avoid spurious regression. Moreover, unit roots in non−stationary series can be eliminated by differencing [51]. However, most of panel unit root tests have statistical limitations in relation to sample size and effect of test [52,53]. In this study, to avoid the contingency of the results of a single panel unit root test, we use the Levin−Lin−Chu (LLC) test [54], IPS test [55], ADF−Fisher Chi−square test, and PP−Fisher Chi−square test [56,57,58]. Proof of panel unit root test is shown in Appendix B.

### 3.6. Panel Cointegration Test

Next, we use Kao’s test [59] to test for cointegration. For panel regression model:(4)$$y_{\mathrm{it}}{=x}_{\mathrm{it}}{\beta +z}_{\mathrm{it}}{\gamma +\varepsilon }_{\mathrm{it}},\;i=1,2,\cdots ,N;\;t=1,2,\cdots ,T$$

In this formula, $y_{\mathrm{it}}{=y}_{i,t-1}{+\mu }_{\mathrm{it}}$, $x_{\mathrm{it}}{=x}_{i,t-1}{+\delta }_{\mathrm{it}}$, $z_{\mathrm{it}}\gamma$ denotes a vector of deterministic components, including constant and linear time trends, $\varepsilon_{\mathrm{it}}$ is the error term.

Kao’s test assumes that there is no cointegration relationship within the variables. Therefore, $\varepsilon_{\mathrm{it}}$ is a non−stationary I (1) process. So, we need to perform a unit root test for the estimated error term. First, we use OLS to estimate Equation (4) and set the residual of Equation (4) as $e_{\mathrm{it}}$. Secondly, we use the ADF test for $e_{\mathrm{it}}$. It is shown as follows:(5)$$e_{\mathrm{it}}{=\rho }_{i}e_{i,t-1}+\overset{p}{\underset{m=1}{\sum }}\gamma_{m}\Delta e_{i,t-m}{+v}_{\mathrm{it}}$$

In which $v_{\mathrm{it}}$ is the error term, ADF statistics $\rho_{i}$ and $\gamma_{m}$ indicate the coefficients of $e_{i,t-1}$ and $\Delta e_{i,t-m}$. ADF test assumes that $H_{0}$ is $\rho_{i}=\rho ,\;\forall_{i}$. Moreover, the ADF statistic is constructed as follows:(6)$$\begin{array}{c} \mathrm{ADF}=\frac{t_{\mathrm{ADF}}+\frac{\sqrt{6N}\sigma_{v}}{{2\sigma }_{0u}}}{\sqrt{\frac{\sigma_{0u}^{2}}{{2\sigma }_{v}^{2}}+\frac{{3\sigma }_{v}^{2}}{{10\sigma }_{0v}^{2}}}}\sim \;N(0,1) \end{array}$$

Among them, $\sigma_{v}^{2}{=\sigma }_{u}^{2}{+\sigma }_{u\varepsilon }^{2}\sigma_{\varepsilon }^{2}$, and $\sigma_{0v}^{2}{=\sigma }_{0\mu }^{2}-\sigma_{0\mu \varepsilon }^{2}\sigma_{0\varepsilon }^{2}$.

### 3.7. Causality Test

Next, we employed the Granger causality test [60] to investigate the causality among agricultural carbon emissions, financial support for agriculture, and chemical fertilizer use. The regression model is shown as follows:(7)$$\begin{array}{c} Y_{i,t}{=c}_{i}+\overset{m}{\underset{k=1}{\sum }}\alpha_{i}X_{i,t-k}+\overset{m}{\underset{k=1}{\sum }}\beta_{i}Y_{i,t-k}{+\mu }_{i,t},\;i=1,2,\cdots ,N;\;t=1,2,\cdots ,T \end{array}$$

$Y_{\mathrm{it}}$ and $X_{\mathrm{it}}$ are the observed values of two stationary variables for the i-th individual in the t-th time. $c_{i}$ is the intercept term; $X_{t-k}$ and $Y_{t-k}$ represent the lag terms of X and Y; $\mu_{\mathrm{it}}$ is the error term; $\alpha_{i}$ and $\beta_{i}$ express the coefficients of $X_{t-k}$ and $Y_{t-k}$. If $\left\{X_{i,t}\right\}$ does not Granger Cause $\left\{Y_{i,t}\right\}$, Equation (7) will change as follows:(8)$$Y_{i,t}=c+\overset{m}{\underset{n=1}{\sum }}\beta_{i}Y_{i,t-n}{+\mu }_{i,t}$$

Next, we can use OLS to estimate Equations (7) and (8), calculate their residual sum of squares, respectively, and compute F-statistic. F-statistic is computed as follows:(9)$$F=\frac{\left({\mathrm{SSR}}_{0}{-\mathrm{SSR}}_{1}\right)\left(N-2n-1\right)}{{n\cdot \mathrm{SSR}}_{1}}$$
where N represents sample size, ${\mathrm{SSR}}_{1}$ and ${\mathrm{SSR}}_{0}$ denote the residual sum of squares of Equations (7) and (8), respectively, n indicates that Equation (8) is restricted to n exclusionary constraints.

### 3.8. Autoregressive Distributed Lag Model (ARDL)

To better understand the dynamic relationship within our interested variables, we employ the ARDL model [61,62]. Compared with other estimations, the ARDL model has three advantages: (1) Firstly, whether integration of order 0 or 1, the ARDL model can estimate the long−run relationship between the variables [63]. (2) Secondly, it is suitable for small data, easy to operate, and will provide sufficient lags [64]. (3) Thirdly, the error correction model (ECM) can be obtained from the ARDL model. We can use ECM to explain the short−run relationship within the variables. The estimated ARDL formula used in this study can be expressed as follows:(10)$$\begin{array}{c} \Delta {\mathrm{lnpercarbon}}_{t}{=\alpha }_{0\;}{+\alpha }_{1}{\mathrm{lnpercarbon}}_{t-1}{+\alpha }_{2}{\mathrm{lnperfertilizer}}_{t-1}{+\alpha }_{3}{\mathrm{lnagriratio}}_{t-1}+\\ \overset{p}{\underset{k=1}{\sum }}\beta_{1}\Delta {\mathrm{lnpercarbon}\;}_{t-k}+\overset{q}{\underset{k=1\;}{\sum }}\beta_{2}\Delta {\mathrm{lnperfertilizer}}_{t-k}+\overset{m}{\underset{k=1}{\sum }}\beta_{3}\Delta {\mathrm{lnagriratio}}_{t-k}+\\ {\omega \mathrm{ECM}}_{t-k}{+\varepsilon }_{t},\;t=1,2,\cdots ,T \end{array}$$
where $\alpha_{1},\;\alpha_{2},\;\alpha_{3}$ represent coefficients in the long run; $\beta_{1}$, $\beta_{2}$, $\beta_{3}$ indicate coefficients in the short run; ${\mathrm{ECM}}_{t-k}$ expresses the error correction term; p, q, and k represent the lag periods of variables, respectively; ω is ECM−coefficients and $\varepsilon_{t}$ is the error term.

### 3.9. FMOLS and DOLS

To ensure the robustness of the outcome in Section 3.8, we further employ two static models, FMOLS and DOLS, to estimate the long−run relationship between carbon emissions from agricultural production, financial support for agriculture, and chemical fertilizer use [65,66]. Compare with other methods, FMOLS is a residual−based test suitable for a small sample size and eliminates sequence correlation and endogeneity among regressors [67]. Moreover, compared with FMOLS, the DOLS estimator has a better sample property in small sample using Monte Carlo simulations and eliminates correlation among regressors [68]. The FMOLS can be explained as given:(11)$${\hat{\beta }}_{\mathrm{GFMOLS}}^{*}=N^{-1}\sum_{i=1}^{N}{\hat{\beta }}_{\mathrm{GFMOLS},i}^{*}$$

In which ${\hat{\beta }}_{\mathrm{GFMOLS}}^{*}$ denotes the i-th panel member’s FMOLS estimator. Moreover, the t-statistic related to it is:(12)$$t_{{\hat{\beta }}_{\mathrm{GFMOLS}}^{*}}=N^{-\frac{1}{2}}\sum_{i=1}^{N}{\hat{\beta }}_{\mathrm{GFMOLS},i}^{*}$$

Then, the DOLS estimator is written as follows:(13)$$\begin{array}{c} y_{\mathrm{it}}{=\beta }_{i}^{\prime }x_{i\;t}+\overset{q}{\underset{j=-q}{\sum }}\delta_{\mathrm{ij}}\Delta x_{i,t+j}{+\varepsilon }_{\mathrm{it}} \end{array}$$

In which q is known as the number of leads/lags for the models. And the estimated coefficient $\beta_{i}^{\prime }$ is expressed as follows:(14)$${\hat{\beta }}_{\mathrm{DOLS}}^{*}=N^{-1}\sum_{i=I}^{N}{\left(\sum_{t=1}^{T}Z_{\mathrm{it}}Z_{\mathrm{it}}^{i}\right)}^{-1}\left(\sum_{t=1}^{T}Z_{\mathrm{it}}\gamma_{\mathrm{it}}^{*}\right)$$
where $Z_{\mathrm{it}}=\left(x_{\mathrm{it}}-\bar{x_{i}},\Delta x_{\mathrm{it}-j},\cdots ,\Delta x_{\mathrm{it}+k}\right)$ is the 2(k + 1) vector of regressors.

### 3.10. Variance Decomposition

Although we discussed the “causality” between carbon emissions from agricultural production, chemical fertilizer use, and financial support for agriculture through the Granger causality test, the outcome of the Granger causality tests only reflects static long−run relationships between variables. However, variance decomposition methods, systematically describing the contribution components of impact changes in each stage, can reflect the dynamic characteristics of VAR model. A VMA (∞) formula is expressed as follows:(15)$$\begin{array}{c} y_{i,t}=\overset{k}{\underset{\;j=1}{\sum }}{(\theta }_{\mathrm{ij}}^{\left(0\right)}\varepsilon_{j,t\;}{+\theta }_{\mathrm{ij}}^{\left(1\right)}\varepsilon_{j,t-1}{+\theta }_{\mathrm{ij}}^{\left(2\right)}\varepsilon_{j,t-2}+\cdots ),\;i=1,2,\cdots ,k,\;t=1,2,\cdots ,T \end{array}$$

Assuming that there is no sequence autocorrelation in $\varepsilon_{j}$. And:(16)$$\begin{array}{c} E\left[{{(\theta }_{\mathrm{ij}}^{\left(0\right)}\varepsilon_{j,t}{+\theta }_{\mathrm{ij}}^{\left(1\right)}\varepsilon_{j,t-1}{+\theta }_{\mathrm{ij}}^{\left(2\right)}\varepsilon_{j,t-2}+\cdots )}^{2}\right]=\overset{\infty }{\underset{q=0}{\sum }}{{(\theta }_{\mathrm{ij}}^{\left(q\right)})}^{2}\sigma_{\mathrm{jj}},\;i,\;j=1,2,\cdots ,k \end{array}$$

The contribution degree of each disturbance term to the variance of $y_{i}$ is measured as follows:(17)$$\begin{array}{c} {\mathrm{RVC}}_{j\to i}\left(\infty \right)=\frac{\sum_{q=0}^{\infty }{{(\theta }_{\mathrm{ij}}^{\left(q\right)})}^{2}\sigma_{\mathrm{jj}}}{\mathrm{var}\left(y_{i}\right)}=\frac{\sum_{q=0}^{\infty }{{(\theta }_{\mathrm{ij}}^{\left(q\right)})}^{2}\sigma_{\mathrm{jj}}}{\sum_{j=1}^{k}\left\{\sum_{q=0}^{\infty }{{(\theta }_{\mathrm{ij}}^{\left(q\right)})}^{2}\sigma_{\mathrm{jj}}\right\}} \end{array}$$

In which RVC is the variance contribution rate used to measure the relative contribution ratio between variables based on impact.

## 4. Results

### 4.1. Cross−Sectional Dependence Tests

Table 3 illustrates the results of cross−sectional dependence tests. We can find that the existence of cross−sectional correlation among variables cannot be rejected at a 1% significance level. The results represent a correlation among different provinces (cities) in China, which may indicate that these provinces (cities) in China have similar behavior in terms of emission from agricultural production.

### 4.2. Unit Root Tests

The results of unit root tests (LLC, IPS, ADF−Fisher, and PP−Fisher tests) are shown in Table 4. The results reveal that all the variables are non−stationary at the 1% significance level, except lnagriratio, which rejected the null hypothesis when only individual effects are considered. Moreover, all the variables were stationary after the first−order difference.

### 4.3. Panel Cointegration Test

Table 5 illustrates the results of Kao’s test (ADF). The results demonstrate that the t−statistic is significant at the 1% level. Therefore, co−integration between the variables exists. The passing of panel cointegration test allowed us to examine the effects of fertilizer and financial support for agriculture on agricultural carbon emissions.

### 4.4. VAR Stability Test

The VAR model of financial support for agriculture, fertilizer, and agricultural carbon emissions is established in this study, and the optimal lag period is 2. Figure 1 represents the inverse roots of the AR characteristic polynomial. All the inverse roots of AR characteristic polynomial are not outside the unit circle, supporting the VAR model’s stability. Hence, we can use variance decomposition to make inferences in a statistical sense.

### 4.5. Benchmark Results

Table 6 shows the results of ARDL in the short−run and the long−run. The variables’ long−run and short−run coefficients are all significant at 5%. The long−term coefficient of lnperfertilizer suggests that agricultural carbon emissions will increase by 1.1713% with a 1% increase in chemical fertilizer use. Moreover, the short−term coefficient of lnperfertilizer implies that a 0.65% increase in agricultural carbon emissions will occur when chemical fertilizer consumption increases by 1%. The results represent that the growth of carbon emission from agricultural production caused by chemical fertilizer use will gradually increase with chemical fertilizer consumption.

In addition, the long−term coefficient of lnagriratio denotes that agricultural carbon emissions will decrease by −0.20% with a 1% increase in financial support for agriculture. Moreover, the short−term coefficient of lnperfertilizer represents that a 0.06% decrease in agricultural carbon emissions will occur when financial support for agriculture increases by 1%.

Finally, the coefficient of the error−correction term expresses that if the agricultural carbon emissions exceed the equilibrium level by 1% in the last year, the agricultural carbon emissions will be corrected by −4.47% this year. The ARDL estimates generally show a long−run and short−run relationship between all variables.

### 4.6. Robustness Check

Table 7 expresses the results of FMOLS and DOLS. The coefficients of chemical fertilizer use in FMOLS and DOLS are 0.9267 and 0.9710, respectively. Furthermore, the coefficients of financial support for agriculture in FMOLS and DOLS are −0.1563 and −0.1713, respectively. Moreover, the above coefficients are all significant at the 1% level. Therefore, we can conclude that financial support for agriculture and agricultural carbon emissions have negative association, and chemical fertilizer use would increase agricultural carbon emissions in a significant way. The results of DOLS and FMOLS confirm the long−term estimation of the ARDL model.

### 4.7. Granger Causality Test

Table 8 shows the Granger causality test’s results on agricultural carbon emissions, chemical fertilizer use, and financial support for agriculture. The results represent one−way causality between financial support for agriculture and agricultural carbon emissions at the 1% significance level. Moreover, the Granger causality test shows the two−way causality between chemical fertilizer consumption and agricultural carbon emissions and the two−way causality between financial support for agriculture and chemical fertilizer use. The results indicate that financial support for agriculture helps to predict carbon emissions from agricultural production. Moreover, financial support for agriculture may have potential for reducing carbon emissions from agricultural production. The Granger causalities between carbon emissions from agricultural production, financial support for agriculture, and chemical fertilizer use are shown in Figure 2 for readers to better understand.

### 4.8. Variance Decomposition

Table 9 shows the variance decomposition results of the 15−year forecast period. In the 15−th period, 96.55% of agricultural carbon emissions can be explained by itself, and the contribution of financial support for agriculture and chemical fertilizer use is 0.22% and 3.22%, respectively. Moreover, the contribution of financial support for agriculture to chemical fertilizer use is 92.36% in the 15−th period. The results illustrate that chemical fertilizer consumption and financial support for agriculture will continue to affect carbon emissions from agricultural production. In addition, financial support for agriculture will be a critical contribution to fertilizer use.

## 5. Conclusions

The main work of this study is to investigate the relationship between agricultural carbon emissions, chemical fertilizer use, and financial support for agriculture by the analysis of panel data for the period 2000–2019 in China. Firstly, we proved the cross−sectional correlation of variables by using the Breusch−Pagan LM test, Pesaran LM test, and Pesaran CD test [49,50]. Then we used unit root tests [54,55,56,57,58] and found that all variables are I (1) process. Next, we found a long−term co−integration relationship between agricultural carbon emissions, chemical fertilizer use, and financial support for agriculture through Kao’s residual panel cointegration test [59]. Based on the above, we established the estimates through the ARDL model [61,62] and found the dynamic relationship between agricultural carbon emissions, chemical fertilizer use, and financial support for agriculture. The results of ARDL illustrate that financial support for agriculture has a significantly negative effect on carbon emissions from agricultural production in the short term and a positive effect in the long term. Next, we employed FMOLS and DOLS to check the robustness of the ARDL estimates, and the robustness of the long−term ARDL estimate is confirmed. In addition, we found two−way causality between chemical fertilizer use and agricultural carbon emissions, two−way causality between chemical fertilizer use and financial support for agriculture, and one−way causality between financial support for agriculture and agricultural carbon emissions through Granger causality tests [60]. Finally, we found that chemical fertilizer consumption and financial support for agriculture will continue to affect agricultural carbon emissions, and chemical fertilizer consumption will be significantly affected by financial support for agriculture. In general, financial support for agriculture have a negative effect on carbon emissions from agricultural production, and financial support for agriculture influences chemical fertilizer use significantly.

The above studies clearly demonstrate the dynamic relationship between financial support for agriculture, chemical fertilizer use, and carbon emissions from agricultural production. We found that financial support for agriculture and chemical fertilizer use has a long−term association. The findings of us are similar to Koondhar et al. [26] and Ismael et al. [69]. However, we have not explored the causal relationship between financial support for agriculture and chemical fertilizer use. In addition, the positive effect of chemical fertilizer use on carbon emissions from agricultural production is also found in this research. The result of our study is similar to the results of Huang et al. [33]. This phenomenon may be due to China has overuse of chemical fertilizers to ensure the grain output. NBSC [70] indicated that since 1990, China’s fertilizer consumption had increased by 103%, but only 50% in return for grain production. The overuse of fertilizer will lead to severe problems such as soil nutrient depletion, soil acidification, nutrient run−off, and reduced biological diversity. These issues caused by the overuse of fertilizer will reduce agricultural production efficiency and lead to more chemical fertilizer use to improve agricultural production efficiency. Moreover, the results of our study express that the increase in financial support for agriculture leads to a decline in carbon emissions from agricultural production. The results are similar to the finding of Liu et al. [31], Han et al. [32], Huang et al. [33], and Chen and Chen [34]. They all found that financial support for agriculture reduced agricultural carbon emissions. The main reason for it is the increase in China’s agricultural production efficiency. China’s agricultural production efficiency continues to improve [71]. When the agricultural production efficiency is at a high level, increasing financial support for agriculture can not only meet the basic capital needs for agricultural production, but also promote the agricultural production structure transiting from traditional agriculture to green agriculture, so as to reduce agricultural carbon emissions.

Based on our findings, there are several policies put forward by us for promoting sustainable agricultural production. Firstly, the government should uphold the concept of sustainable agriculture, increase financial support for environmental−friendly agriculture, and encourage the research and use of cleaner agricultural production technologies and chemical fertilizer substitutes to ensure that agriculture, especially in undeveloped areas, promotes economic development with less carbon emissions. Secondly, the government should stop farmers from blindly investing in agricultural production factors, especially chemical fertilizer, for high yield by strengthening the awareness of farmers’ environmental production, improving the carbon tax system, and implementing reward and punishment measures for agricultural carbon emissions. Thirdly, the government, especially those of developing countries, can establish cooperative relations with the countries with developed sustainable agriculture and introduce advanced low−carbon management technologies and systems from them.

This study provides substantial evidence for reducing agricultural carbon emissions, fills the research gap in financial support for agriculture’s environmental effect, and riches the research on carbon emissions, especially in agriculture, based on previous research. However, our study still has some deficiencies: this study uses small dataset, which is a restriction of the study, although methods suitable for small dataset is used in this study. Moreover, the research lacks research on the mechanism of financial support for agriculture affecting carbon emissions from agricultural production. It only considers the impact of financial support for agriculture on chemical fertilizer use. In addition, due to the differences in different parts of China, there is a spatial effect in the impact of financial support for agriculture on agricultural carbon emissions [32]. We ignore it in this study. Therefore, we suggest that future research can consider the spatial effect of financial support for agriculture on carbon emissions from agricultural production and focus on the mechanism of financial support for agriculture affecting carbon emissions from agricultural production to implement targeted financial support for agriculture for different regions to reduce carbon emissions from agricultural production.

## Figures and Tables

**Figure 1 ijerph-19-07155-f001:**
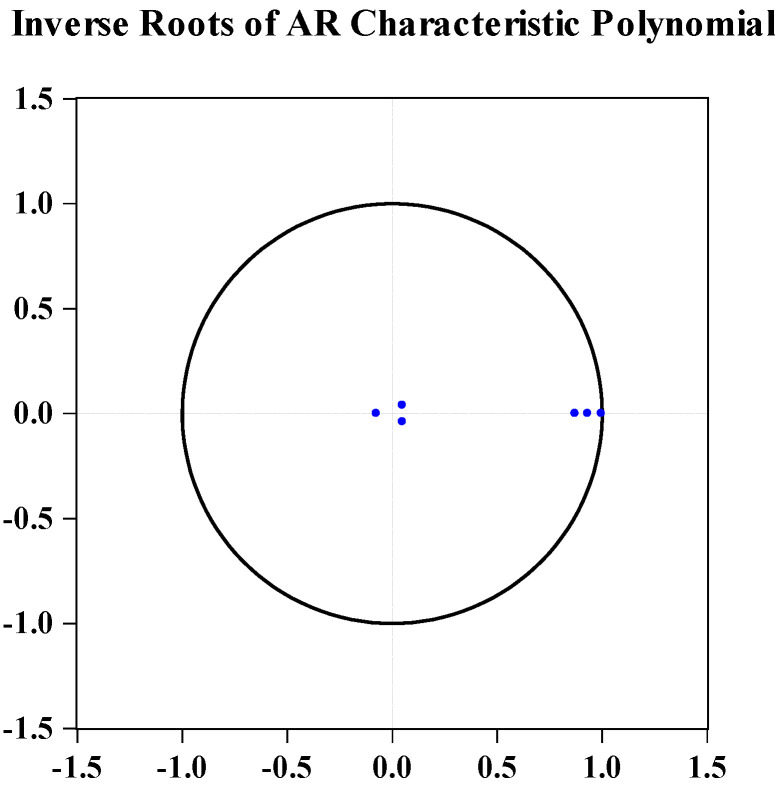
The results of the stability test.

**Figure 2 ijerph-19-07155-f002:**
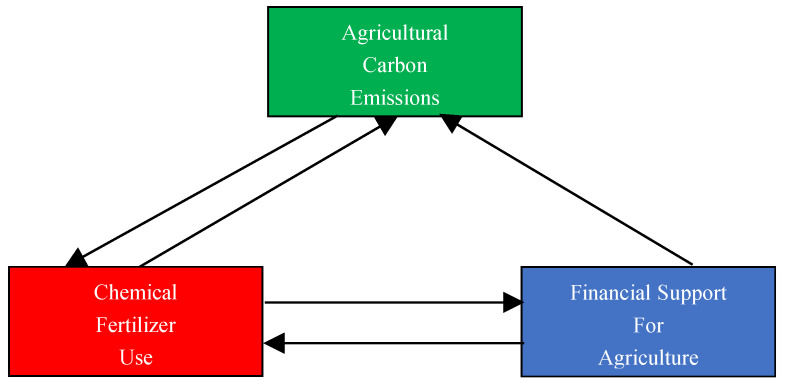
The Granger causalities between variables. A → B represents A Granger cause B. For example, financial support for agriculture → agricultural carbon emissions means that there is a one−way causality between chemical fertilizer use and agricultural carbon emissions.

**Table 1 ijerph-19-07155-t001:** Carbon emission coefficient reference.

Carbon Source	Carbon Emission Coefficient	Reference
Fertilizer	0.8956 kg/kg	Oak Ridge National Laboratory [45]
Pesticides	4.9341 kg/kg	Oak Ridge National Laboratory
Mulches	5.18 kg/kg	Institute of Resource, Ecosystem, and Environment of Agriculture, Nanjing Agricultural University [13]
Diesel	0.5927 kg/kg	Intergovernmental Panel on Climate Change IPCC [13]
Plowing	312.6 kg/hm^2^	College of Biological Sciences, China Agricultural University
Irrigation	25 kg/hm^2^	[46]
Pigs	34.0910 kg/(each·year)	Intergovernmental Panel on Climate Change IPCC [13]
Cattle	415.91 kg/(each·year)	Intergovernmental Panel on Climate Change IPCC [13]
Sheep	35.1819 kg/(each·year)	Intergovernmental Panel on Climate Change IPCC [13]
Agricultural electricity	CO_2_: 0.7921 t·MWh^−1^	China’s Ministry of Ecology and Environment

**Table 2 ijerph-19-07155-t002:** The definition and measurement of variables.

Variables	Definition	Measurement
Agricultural carbon emissions (percarbon)	Average carbon emissions from agricultural production	Total agricultural carbon emissions /Arable land area
chemical fertilizer use (perfertilizer)	Average chemical fertilizer consumption in agricultural production	Total chemical fertilizer consumption /Arable land area
financial support for agriculture (agriratio)	The ratio of agriculture, forestry, and water in financial expenditure	Total financial expenditure of agriculture, forestry, and water/Total financial expenditure

**Table 3 ijerph-19-07155-t003:** The results of cross−sectional dependence tests.

Test	Statistic	Prob.
Breusch−Pagan LM	2281.3470	0.0000
Pesaran scaled LM	62.5970	0.0000
Pesaran CD	7.2394	0.0000

**Table 4 ijerph-19-07155-t004:** The results of unit root tests.

Variables	Level			First−Difference		
	Intercept	Intercept and Trend	Conclusion	None	Intercept and Trend	Conclusion
LLC test						
Lnpercarbon	0.6898	0.9999	N	0.0000	0.0000	S
Lnperfertilizer	0.0000	1.0000	U	0.0000	0.0000	S
Lnagriratio	0.0000	0.7712	U	0.0000	0.0000	S
IPS test						
Lnpercarbon	0.9691	1.0000	N	0.0000	0.0000	S
Lnperfertilizer	0.1394	1.0000	N	0.0000	0.0000	S
Lnagriratio	0.0000	1.0000	U	0.0000	0.0000	S
ADF−Fisher Chi−square test						
Lnpercarbon	0.9037	0.9564	N	0.0000	0.0000	S
Lnperfertilizer	0.2152	1.0000	N	0.0000	0.0000	S
Lnagriratio	0.0000	1.0000	U	0.0000	0.0000	S
PP−Fisher Chi−square test						
Lnpercarbon	0.9539	1.0000	N	0.0000	0.0000	S
Lnperfertilizer	0.0194	1.0000	N	0.0000	0.0000	S
Lnagriratio	0.3108	1.0000	N	0.0000	0.0000	S

Note: U indicates Unknown, N indicates non−stationary, and S indicates stationary.

**Table 5 ijerph-19-07155-t005:** The results of Kao’s test (ADF).

	Null Hypothesis	t−Statistics	Prob.
ADF	No co−integration	−6.523558	0.0000

**Table 6 ijerph-19-07155-t006:** The results of ARDL analysis.

Variable	Coefficient	Std.Error	t−Statistic	Prob.
	Long Run Equation			
Lnperfertilizer	1.1713	0.0392	29.9010	0.0000
Lnagriratio	−0.2015	0.0550	3.6665	0.0003
	Short Run Equation			
CointeQ01	−0.0447	0.0173	−2.5799	0.0103
D(Lnperfertilizer)	0.6500	0.0649	10.0108	0.0000
D(Lnagriratio)	−0.0551	0.0148	−3.7361	0.0002
C	0.0484	0.0212	2.2830	0.0230

**Table 7 ijerph-19-07155-t007:** The results of the robustness check.

Variable	Coefficient	Std.Error	t−Statistic	Prob.
FMOLS				
Lnperfertilizer	0.9267	0.0230	40.2135	0.0000
Lnagriratio	−0.1563	0.0125	−12.5067	0.0000
DOLS				
Lnperfertilizer	0.9710	0.0214	45.3548	0.0000
Lnagriratio	−0.1713	0.0134	−12.7897	0.0000

**Table 8 ijerph-19-07155-t008:** Pairwise Granger causality tests.

Null Hypothesis:	F−Statistic	Prob.
LNPERFERTILIZER does not Granger Cause LNPERCARBON	6.4082	0.0000
LNPERCARBON does not Granger Cause LNPERFERTILIZER	8.2025	0.0000
LNAGRIRATIO does not Granger Cause LNPERCARBON	6.7647	0.0000
LNPERCARBON does not Granger Cause LNAGRIRATIO	0.8131	0.6164
LNAGRIRATIO does not Granger Cause LNPERFERTILIZER	2.6765	0.0044
LNPERFERTILIZER does not Granger Cause LNAGRIRATIO	2.6333	0.0050

**Table 9 ijerph-19-07155-t009:** Variance decomposition results.

Period	S.E.	Lnpercarbon	Lnperfertilizer	Lnagriratio
Variance Decomposition of Lnpercarbon:				
1	0.15	100.00	0.00	0.00
2	0.20	99.63	0.09	0.27
3	0.23	99.52	0.18	0.29
4	0.26	99.40	0.32	0.28
5	0.27	99.24	0.50	0.26
6	0.29	99.05	0.71	0.24
7	0.30	98.82	0.95	0.23
8	0.30	98.57	1.21	0.22
9	0.31	98.30	1.49	0.21
10	0.31	98.02	1.77	0.21
11	0.32	97.72	2.07	0.21
12	0.32	97.43	2.36	0.21
13	0.32	97.13	2.65	0.21
14	0.32	96.84	2.94	0.22
15	0.33	96.55	3.22	0.22
Variance Decomposition of Lnperfertilizer:				
1	0.15	88.81	11.19	0.00
2	0.20	88.18	11.70	0.13
3	0.24	87.60	12.29	0.11
4	0.26	87.15	12.76	0.09
5	0.27	86.72	13.20	0.09
6	0.29	86.28	13.61	0.10
7	0.30	85.84	14.02	0.14
8	0.30	85.39	14.42	0.19
9	0.31	84.92	14.82	0.26
10	0.31	84.44	15.22	0.35
11	0.31	83.95	15.61	0.44
12	0.32	83.46	15.99	0.55
13	0.32	82.97	16.37	0.67
14	0.32	82.46	16.75	0.79
15	0.32	81.95	17.12	0.92
Variance Decomposition of Lnagriratio:				
1	0.23	0.67	20.86	78.47
2	0.31	0.79	27.80	71.41
3	0.36	0.88	29.15	69.97
4	0.40	0.93	29.19	69.88
5	0.43	0.96	28.71	70.32
6	0.46	0.98	28.01	71.00
7	0.48	1.00	27.21	71.79
8	0.49	1.00	26.37	72.62
9	0.51	1.00	25.54	73.45
10	0.52	1.00	24.74	74.26
11	0.53	0.99	23.99	75.02
12	0.54	0.98	23.30	75.72
13	0.55	0.97	22.67	76.36
14	0.55	0.95	22.12	76.93
15	0.56	0.94	21.65	77.42

## Data Availability

Data supporting the conclusions of this article are included within the article. The dataset presented in this study are available on request from the corresponding author.

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
