# Peer review of "Financial Support for Agriculture, Chemical Fertilizer Use, and Carbon Emissions from Agricultural Production in China"

_ijerph, 2022, doi:10.3390/ijerph19127155_

Round 1
Reviewer 1 Report
Dear Authors,
Please see my comments attached.
Best Regards,

Author Response
First of all, we would like to thank Reviewer 1 for reading our article and for your valuable comments. Below is our response to all comments made.
Line 15: I don’t think you should present these methods in the Abstract. If somebody is not
familiar with the cross-sectional or time series methods, ARDL, DOLS etc. does not say too
much.
Response 1: we accepted the reviewer's suggestion not to present these methods in the Abstract, and used “a series of methods” to replace the names of all the methods we used in this study.
Line 40: “Unfortunately, it has consumed a lot of agricultural materials,” What has consumed a lot of materials (and what materials, exactly?)
Response 2: We are sorry our expression is not clear. Following the reference, there has been corrected as “agricultural production has consumed a lot of agricultural materials, such as fertilizers, fuels, pesticides, and feeds, which are essential sources of carbon emissions from agricultural production.”
Line 43: “agricultural sector” Agricultural sector or food system? (furthermore, with or without land use change)
Response 3: ‘20% of China's total greenhouse gas emissions’ do not totally come from food system, and are not accompanied by land use change. To better follow the reference, we changed “agricultural sector” to “agricultural activities”.
Line 51: „chemical fertilizer has become an important measure 51 to alleviate the pressure
on agriculture” Do you mean that decreasing the level of chemical fertilizer used? This sentence is not entirely clear.
Response 4: We are sorry that our expression has caused the respected reviewer misunderstanding. This sentence has been corrected as “With limited resources, lack of cultivated land, and rapid population growth, chemical fertilizer, as an agricultural material that helps to improve yield, has become an important measure to alleviate the pressure on agriculture [7]”.
Table1: In the Carbon Emission Coefficient column, first two rows (0.895 6 kg/kg and 4.934 1 kg/kg), there is an unnecessary space before the last digit (I believe).
Response 5: We sincerely accept your suggestion to delete these two unnecessary spaces.
Table1: There is no reference to Intergovernmental Panel on Climate Change IPCC and
China’s Ministry of Ecology and Environment. Furthermore, you should unify the use of
capital letter in case of the first column.
Response 6: We have added references to Intergovernmental Panel on Climate Change IPCC and China’s Ministry of Ecology and Environment and unify the use of capital letter in case of the first column.
Line 171: You should provide references for the sources.
Response 7: we accepted the reviewer's suggestion to provide references for the sources as follows:” The data are collected from the China Rural Statistical Yearbook (2001-2020), China Statistical Yearbook (2001-2020), the China National Bureau of Statistics, and the China Population and Employment Statistics Yearbook (2001-2020)”.
Line 175: “To avoid heteroscedasticity, we process the data logarithmically 175 for the analysis.” Log transformation does not provide a full protection against heteroscedasticity, it is rather helpful to ensure linearity among the variables. I think you should write “mitigate” instead of “avoid”.
Response 8: we accepted the reviewer's suggestion to write ”mitigate” instead of “avoid”.
Line 187, equation 3: The lnagriratio beta should be Beta2 instead of Beta1.
Response 9: We have corrected the coefficient of lnagriratio as Beta2.
Line 187, equation 3: Is it necessary the (1) subscript for the constant? Is the constant is the
same across provinces and time periods?
Response 10: The subscript for the constant is unnecessary. We delete it.
3.4 Cross-sectional Dependence test: I think this section can be placed in the Appendix,
because it is not the main interest of this study. Furthermore, you should define the Chi-square
distribution notation and the Normal distribution (especially the latter, because the number of
data points was denoted by N as well).
Response 11: Cross-sectional dependence test is a pre-test for out study, and it is not the main interest of our study. This section has been placed in the Appendix A. And we also have added the definition of the Chi-square distribution notation and the Normal distribution.
Line 211: “Spurious regression leads to poor effectiveness of the estimation results.” I think poor effectiveness is not the correct wording here: it leads to incorrect estimates and will most likely indicate a non-existing relationships.
Response 12: The new expression suggested by the reviewers is indeed more concise and academic. We have corrected this sentence as ” Spurious regression leads to incorrect estimates and will most likely indicate a non-existing relationships”.
Line 211-212: “The panel 211 unit root test is necessary for the panel to avoid spurious regression. Because it makes the 212 variable series stationary by differencing them [49].” This should be rewritten. Panel unit root test is necessary to establish the presence of unit root. Non-stationary series contains unit root, which can be removed by differencing (in case of unit root non-stationarity). I believe you wrote this, but your line of reasoning is not entirely correct.
Response 13: As the respected reviewer said, our line of reasoning is not entirely correct. We accepted the reviewer's suggestion to correct this sentence as follows: “The panel unit root test is necessary for the panel to avoid spurious regression. Moreover, unit roots in non-stationary series can be eliminated by differencing [52]”.
Line 215: What is the LLC Test? Is it a shortcut? This should be placed in the Appendix as
well.
Response 14: The LLC test is the shortcut of Levin-Lin-Chu test. This has been placed in the Appendix B.1
Line 219: What is “m” in this test? The subscript for alpha and d is “mt”.
Response 15: According to Levin et al., (2002). “m” has three values, 1,2,3. is used to indicate the vector of deterministic variables and is used to indicate the corresponding vector of coefficients for a particular model. m = 1; 2; 3. Thus, = ∅ (the empty set); = {1} and = {1; t}. We added “m=1,2,3” and the explanation of in our article.
3.5.2 IPS Test and 3.5.3. ADF-Fisher and PP-Fisher Tests. Place them in the Appendix
please. These are just general pre-tests necessary for frequentist inference, but they are not the
main interest.
Response 16: We have placed them in the Appendix B.2 and B.3.
Line 256: „Firstly, the ADF 256 and PP tests are performed on each individual.” On each
individual what?
Response 17: Each individual in this study indicates 31 provinces (cities) of China. We have corrected this sentence as “The ADF and PP tests are performed on the time series of each individual”.
Line 259: „ADF-Fisher 259 and PP-Fisher tests assume that H0 is the root of existence
unit.” This sentence is not clear grammatically. I think it should be something simpler (the
associated H0 assumes the presence of unit root for example).
Response 18: The new expression suggested by the reviewers is indeed more concise in grammar. We have corrected it as “ADF-Fisher and PP-Fisher tests assume that the existence of unit root”.
Line 270: „Therefore, is a non-cointegration I (1) process.” I think you mean non-
stationary I(1) process.
Response 19: We have corrected it as “Therefore, is a non-stationary I (1) process”.
Line 275:You write Δei,t-p, but in the equation this part has the running index of (t-m). Is
this correct?
Response 20: We are sorry for our negligence. We have corrected Δei,t-p as
3.7. VAR Stability Check. This is usually not reported in details. I think it suffices to report that whether the VAR is stable or not (or equivalently, report the estimated eigenvalues).
Response 21: We have deleted section 3.7 and only reported the outcome of VAR stability check in section 4.4.
Line 308: “Firstly, whether the variables are of the same order, the ARDL model”. Do you mean whether or not the variables are of the same order?
Response 22: We are sorry that our expression is not entirely correct, we want to express that whether integration of order 0 or 1, the ARDL model can estimate the long-run relationship between the variables.
Line 309: “Secondly, it reduces the 309 panel data’s “non-stationary” problem.” How does
it reduce?
Response 23: We want to explain that no matter whether the variables are I(0) or I(1), ARDL can estimate the long-term relationship, which solves the nonstationary problem related to time series in a certain way. Actually, this is part of the first advantage of ARDL. To make our expression more concise, we have deleted it.
Line 324: “regressors. [61].” The dot is misplaced.
Response 24: We have deleted this unnecessary dot.
Line 325: “DOLS has a better estimate and eliminates correlation among regressors” What
does it mean to have a „better estimate”?
Response 25:According to Kao and Chiang (2001), compared with FMOLS, the DOLS estimator has a better sample property in small sample using Monte Carlo simulations. We have used “a better sample property in small sample using Monte Carlo simulations” to replace “have a better estimate”.
Line 338 – 343: This is incorrect. You imply that since Granger causality is not true causality (which is entirely true), you use variance decomposition. But this does not provide any information about causality. Furthermore, you write “can solve the 342 problem of the Granger causality test’s result, which is only a statistical estimate. [63].” Variance decomposition is just a „statistical estimate” too, nothing more, nothing less. This part should be corrected.
Response 26:We sincerely accept your suggestion. Actually, the variance decomposition is not a method to explore the casual relationship within variables. We have corrected our expressions as follows:
“Although we discussed the “causality” between carbon emissions from agricultural production, chemical fertilizer use, and financial support for agriculture through the Granger causality test, the outcome of the Granger causality tests only reflects static long-run relationships between variables. However, variance decomposition methods, systematically describing the contribution components of impact changes in each stage, can reflect the dynamic characteristics of VAR model.”.
Line 346: You write: “sequence correlation”. Do you mean autocorrelation?
Response 27: We have corrected “sequence correlation” as “sequence autocorrelation”.
Line 356: “is supported at a 1% significance” I think it would be better to write „cannot reject”, because the word „supported” is not entirely correct to use in statistics (we always have
some uncertainty).
Response 28: The new expression suggested by the reviewers is indeed more concise and academic. We accepted your suggestion to correct this sentence as “We can find that the existence of cross-sectional correlation among variables cannot be rejected at a 1% significance level.”.
Line 357: “The results represent a correlation among different provinces (cities) in China.”
Please provide some more information here (1 sentence). Does this mean that these cities have
similar behavior in terms of emission, for example?
Response 29: Providing some information here is beneficial for us to explain our outcome. We added “which may indicate that these provinces (cities) in China have similar behavior in terms of emission from agricultural production” after “The results represent a correlation among different provinces (cities) in China.”
Line 364: In line 362, you write that “lnagriratio,” is stationary, but in line 364 you write
that it is I(1), non-stationary process. Furthermore, if it is stationary, then you should not
difference it. These results looks quite mixed for me, so the word „stationary” and „non-
stationary” should be highlighted in an additional column.
Response 30: Only individual effects are considered, “lnagriratio” is stationary through LLC test, IPS test, and ADF-Fisher test. In other words, we do not have sufficient evidence to ensure “lnagriratio” is a I(0) process. Moreover, all the variables are stationary through the four tests whether only considering intercept effects or considering intercept and trend effects. In general, we have reason to believe that all the variables are I(1) process. In addition, we accept your suggestion to highlight “stationary” and “non-stationary” in an additional column.
Line 370: “co-integration between the variables is 370 proved.”. Cointegration cannot be
rejected by the test. However, the word „proved” is not correct to use.
Response 31: We agree that your suggestion is correct. And we correct this sentence as “Therefore, co-integration between the variables exists.
Line 371: “The existence of panel cointegration causal link between”. We don’t know
anything about causality here!
Response 32: We only know the cointegration relationship between our interested variables. And this sentence is corrected as” The passing of panel cointegration test allowed us to examine the effects of fertilizer and financial support for agriculture on agricultural carbon emissions”.
Line 377: “the optimal lag period is 15.”. 15 lag means 15 years, in case of annual data. I
don’t think this is correct, it seems extremely large.
Response 33: This is our negligence. The optimal lag period of VAR stability check is 2.
Figure 1. The results of the stability test. You wrote all eigenvalues lies within the unit circle, but this is not true. There are points on the circle or very close to the circle. Please re-
check.
Response 34: Our expression is not entirely consistent with the result. We have corrected “lies within” as “are not outside”.
Line 396: “They will reduce agricultural” Who are you referring by „they”?
Response 35: “They” is “These issues caused by the overuse of fertilizer”.
Line 399 and 400: Please use two decimal places.
Response 36: We accept your suggestion to use two decimal places.
Line 402: “This study proves”. Rather, „This study supports..”
Response 37: We accept the respected reviewer’s suggestion to correct “proves” as “supports”.
Line 414: “financial support for agriculture could significantly reduce carbon”. You should
write that financial support was associated with reducing carbon emission. From the model, we don’t know that financial support actually cause carbon emission to decrease, only that they
have negative association.
Response 38: We sincerely accept your suggestion to correct this expression as “Therefore, we can conclude that financial support for agriculture and agricultural carbon emissions have negative association”.
4.7. Granger Causality Test. Granger causality is about predictability. So if X Granger
cause Y, it means only that X helps to predict Y. Your interpretation involves „increase” and
„decrease”, which is not correct.
Response 39: We sincerely accept your explanation of Granger causality test. This section has been corrected as follows:
“Table 8 shows the Granger causality test’s results on agricultural carbon emissions, chemical fertilizer use, and financial support for agriculture. The results represent one-way causality between financial support for agriculture and agricultural carbon emissions at the 1% significance level. Moreover, the Granger causality test shows the two-way causality between chemical fertilizer consumption and agricultural carbon emissions and the two-way causality between financial support for agriculture and chemical fertilizer use. The results indicate that financial support for agriculture helps to predict carbon emissions from agricultural production. Moreover, financial support for agriculture may have potential for reducing carbon emissions from agricultural production. The Granger causalities between carbon emissions from agricultural production, financial support for agriculture, and chemical fertilizer use are shown in Fig 2 for readers to better understand.”.
Line 426: “The results of this study represent that financial support for agriculture causes
increases 426 in agricultural carbon emissions and chemical fertilizer consumption.” Isn’t this
results says the complete opposite of other results? That financial support was associated with
lower carbon emission? Otherwise, I don’t think it is correct explanation, please see my
previous comment.
Response 40: Section 3.8 has been corrected as follows:
“Table 8 shows the Granger causality test’s results on agricultural carbon emissions, chemical fertilizer use, and financial support for agriculture. The results represent one-way causality between financial support for agriculture and agricultural carbon emissions at the 1% significance level. Moreover, the Granger causality test shows the two-way causality between chemical fertilizer consumption and agricultural carbon emissions and the two-way causality between financial support for agriculture and chemical fertilizer use. The results indicate that financial support for agriculture helps to predict carbon emissions from agricultural production. Moreover, financial support for agriculture may have potential for reducing carbon emissions from agricultural production. The Granger causalities between carbon emissions from agricultural production, financial support for agriculture, and chemical fertilizer use are shown in Fig 2 for readers to better understand.”.
Table 9. Variance decomposition results. It would be sufficient to use only two decimal
places.
Response 41: All the data in table 9 have been kept to two decimal places.
- Conclusion and Policy Implications. Please correct this section accordingly. I think you
should mention, that small dataset is a restriction of the study as well, even if you use methods
suitable for small samples.
Response 42: We accecpt your suggestion to add “this study uses small dataset, which is a restriction of the study, although methods suitable for small dataset is used in this study.” In Section 5.
Reference
- Levin, A.; Lin, C.; James Chu, C., Unit root tests in panel data: Asymptotic and finite-sample properties. Econometrics 2002, 108, 1-24. [CrossRef]

Reviewer 2 Report
Dear authors
This paper is of interesting title and I believe that can certainly grab the attention of international readers in different field of study including environmental, agricultural, and geographical studies. It is mainly focused on understanding the relationship between chemical fertilizer use, financial support for agriculture, and agricultural carbon emissions. These three factors have very significant role in sustainable/unsustainable agricultural production. The authors have used the panel data of 30 provinces (cities) in China from 2000 to 2019 and employed cross sectional dependence tests, unit root tests, panel cointegration test, ARDL, FMOLS, DOLS, and variance decomposition. In other words, these data have been used to analyze the relationship among these three key factors. The methodology of the manuscript is sound and understandable and have created reliable results in my opinion. However, there are some rooms for further improvements in my point of view that should be addressed by the respected authors before acceptance. My main comments are as follows:
- Please mention one of the main applicable recommendations of the study in the end of abstract.
- The introduction section is short and well-organized. However, I believe that the study should be contextualized in this section. In other words, please try to highlight your reasons for choosing China as the main research area. Of course, it should be mentioned that I understand that the respected authors are originally based in China. However, justification of the study area and providing reasons for its selection is an undeniable part of such studies.
- Please define some sub-objectives for the study in the end of introduction section.
- I believe that the authors should try to highlight the main contribution of their study to the body of knowledge in the end of literature review. This point should also be highlighted in the conclusion section.
- Material and methods section has been written very well. However, some economic models have mentioned in this section that their selection should be justified. In other words, what are the main superiorities of these models to similar competing models?
- Please explain the method that you have employed for testing the stability of your results.
- A discussion section should be added to the manuscript in my point of view. Although some the content of the discussion section has been presented in the results section, the authors can enrich it with comparing their results to the results of other researchers (in discussion section).
- Please try to put your recommendations in an international scope. This can improve the attractiveness of your manuscript.
- The main take-home message of the research should be mentioned in the end of conclusion section.
In general, this paper is interesting in my opinion and can be considered for publication in International Journal of Environmental Research and Public Health after major revisions.
Good luck
Author Response
Response to Reviewer 2 Comments
First of all, we would like to thank Reviewer 2 for reading our article and for your valuable comments. Below is our response to all comments made.
- Please mention one of the main applicable recommendations of the study in the end
of abstract.
Response 1: Mentioning one of the main applicable recommendations of the study in the end of abstract is beneficial for readers to quickly obtain the key information of this article. Therefore, we accept your suggestion to add “The government should uphold the concept of sustainable agriculture, increase financial support for environmental-friendly agriculture, and encourage the research and use of cleaner agricultural production technologies and chemical fertilizer substitutes.” in the end of abstract.
2.The introduction section is short and well-organized. However, I believe that the study
should be contextualized in this section. In other words, please try to highlight your reasons for choosing China as the main research area. Of course, it should be mentioned that I understand that the respected authors are originally based in China. However, justification of the study area and providing reasons for its selection is an undeniable part of such studies.
Response 2: We sincerely accept your valuable suggestion. We demonstrate the reasons why we chose China as the main research area of this article as follows:
“EKC hypothesis shows that when a country's economic development level is low, environmental pollution is not serious, with the increase of per capita income, environmental pollution tends from low to high, but decreases per capita income as increases further. The economic level of developing countries is low. Moreover, economic growth is preferred than environmental protection in developing countries [2]. Therefore, environmental pollution in developing countries is a major issue in the world. China is the worlds’ largest developing country with the largest population. Actually, in the past 15 years, China has emitted the most carbon dioxide globally [3]. Therefore, China can be the focus of research on environmental pollution in developing countries.”
- Please define some sub-objectives for the study in the end of introduction section.
Response 3: We define measuring annual agricultural carbon emissions of 31 provinces (cities) in China as our sub-objectives for our study.
- I believe that the authors should try to highlight the main contribution of their study to the body of knowledge in the end of literature review. This point should also be highlighted in the conclusion section.
Response 4: We sincerely accept your valuable suggestion. Therefore, we added “Compared with the existing research, this study discusses the effect of financial support for agriculture on environment at an overall level, fills the research gap in financial support for agriculture’s environmental effect, and riches the research on carbon emissions, especially in agriculture” in the end of literature review. Moreover, “This study provides substantial evidence for reducing agricultural carbon emissions, fills the research gap in financial support for agriculture’s environmental effect, and riches the research on carbon emissions, especially in agriculture, based on previous research” is added in the conclusion section.
- Material and methods section has been written very well. However, some economic models have mentioned in this section that their selection should be justified. In other words, what are the main superiorities of these models to similar competing models?
Response 5: We sincerely accept your suggestion on the advantages of adding the selected model compared with similar competing models. Therefore, we explain the advantages of ARDL, FMOLS, DOLS, and variance decomposition.
ARDL: “Compared with other estimations, the ARDL model has three advantages: (1) Firstly, whether integration of order 0 or 1, the ARDL model can estimate the long-run relationship between the variables [64]. (2) Secondly, it is suitable for small data, easy to operate, and will provide sufficient lags [65]. (3) Thirdly, the error correction model (ECM) can be obtained from the ARDL model.”.
FMOLS and DOLS: “Compare with other methods, FMOLS is a residual-based test suitable for a small sample size and eliminates sequence correlation and endogeneity among regressors [68]. Moreover, compared with FMOLS, the DOLS estimator has a better sample property in small sample using Monte Carlo simulations and eliminates correlation among regressors [69].”
Variance decomposition:” However, variance decomposition methods, systematically describing the contribution components of impact changes in each stage, can reflect the dynamic characteristics of VAR model.”
- Please explain the method that you have employed for testing the stability of your results.
Response 6: We employed FMOLS and DOLS to testing the stability of the long-run outcome of ARDL. And the introduction of FMOLS and DOLS is shown in section 3.9.
- A discussion section should be added to the manuscript in my point of view. Although some the content of the discussion section has been presented in the results section, the authors can enrich it with comparing their results to the results of other researchers (in discussion section).
Response 7: We accept your suggestion to add a discussion in section 5. The discussion is as follows:
“The above studies clearly demonstrate the dynamic relationship between financial support for agriculture, chemical fertilizer use, and carbon emissions from agricultural production. We found that financial support for agriculture and chemical fertilizer use has a long-term association. The findings of us are similar to Koondhar et al. [27] and Ismael et al. [70]. However, we have not explored the causal relationship between financial support for agriculture and chemical fertilizer use. In addition, the positive effect of chemical fertilizer use on carbon emissions from agricultural production is also found in this research. The result of our study is similar to the results of Huang et al. [34]. This phenomenon may be due to China has overuse of chemical fertilizers to ensure the grain output. NBSC [72] indicated that since 1990, China’s fertilizer consumption had increased by 103%, but only 50% in return for grain production. The overuse of fertilizer will lead to severe problems such as soil nutrient depletion, soil acidification, nutrient run-off, and reduced biological diversity. These issues caused by the overuse of fertilizer will reduce agricultural production efficiency and lead to more chemical fertilizer use to improve agricultural production efficiency. Moreover, the results of our study express that the increase in financial support for agriculture leads to a decline in carbon emissions from agricultural production. The results are similar to the finding of Liu et al. [32], Han et al. [33], Huang et al. [34], and Chen and Chen [35]. They all found that financial support for agriculture reduced agricultural carbon emissions. The main reason for it is the increase in China’s agricultural production efficiency. China’s agricultural production efficiency continues to improve [73]. When the agricultural production efficiency is at a high level, increasing financial support for agriculture can not only meet the basic capital needs for agricultural production, but also promote the agricultural production structure transiting from traditional agriculture to green agriculture, so as to reduce agricultural carbon emissions.”
- Please try to put your recommendations in an international scope. This can improve the attractiveness of your manuscript.
Response 8: We emphasize that the policy recommendations in this study are more targeted at sustainable agriculture in developing countries than just China.
- The main take-home message of the research should be mentioned in the end of
conclusion section.
Response 9: We added “In general, financial support for agriculture have a negative effect on carbon emissions from agricultural production, and financial support for agriculture influences chemical fertilizer use significantly” in the end of conclusion.

Round 2
Reviewer 1 Report
Dear Authors,
Thank you for the corrections. I suggest to publish the paper.
Yours sincerely,
Reviewer 2 Report
Dear authors
Thank you so much for your efforts in addressing my comments. I am now satisfied with you responses and revision in the manuscript. Therefore, there is no need for further revisions in my opinion.
Good luck